## Classics

auxin; cytokinin; in vitro propagation; organ patterning; regeneration.

**Corresponding author:**
Charles W. Melnyk;
Email: charles.melnyk@slu.se

# Quantitative regeneration: Skoog and Miller revisited

Charles W. Melnyk

Department of Plant Biology, Linnean Center for Plant Biology, Swedish University of Agricultural Sciences, Uppsala, Sweden

## Abstract

In 1957, Skoog and Miller published their seminal work on the effects of hormones upon plant growth. By varying the concentrations of auxin and cytokinin, they observed dramatic differences in shoot and root growth from tobacco stem cultures. Their finding that quantitative differences in hormone concentrations could dramatically alter the fate of developing organs provided a foundation for understanding organ formation and tissue regeneration. Their in vitro assays established plant propagation techniques that were critical for regenerating transgenic plants. Here, I discuss their original paper, what led to their findings and its impact on our understanding of hormone interactions, how plants regenerate and in vitro tissue culture techniques.

## 1. Introduction

It has been over 65 years since Skoog and Miller published their seminal work on the chemical regulation of organ growth in tissue culture (Skoog & Miller, 1957). After so many years, their paper remains highly cited and relevant to multiple aspects of plant biology. Although this paper is perhaps the best known and most cited from Skoog's laboratory, Folke Skoog had a long history of studying plant growth substances. Originally from Sweden, Skoog earned his undergraduate and PhD degrees at Caltech where he worked on auxin physiology. Later joining the University of Wisconsin-Madison as a faculty member in 1947, he published over 170 papers during his career that largely focused on phytohormones (Armstrong, 2002). Several years after starting in Wisconsin, his lab recruited a postdoctoral associate, Carlos Miller, to continue working on hormone physiology. Miller had an ambitious task, to identify the substance(s) responsible for cell divisions in plant tissue. These years leading up to the 1957 paper showed enormous growth and strong enthusiasm for hormone biology with plant physiologists searching for new factors and characterising recently identified ones (Thimann, 1974). In vitro techniques had been previously established and the role of auxin was being studied intensely. Miller succeeded to identify compounds that promoted cell division, and together with previous work on auxin and in vitro techniques, these formed the basis for the 1957 paper with Skoog. Here, I discuss the background, the paper and the implications that stemmed from the seminal work of Skoog and Miller.

## 2. Establishing de novo organ formation

The ability of plants to form organs and modify their development in response to the environment has been an intense research focus for over a century. However, fundamental to our understanding of such developmental plasticity has been the discovery of the growth hormone auxin (Went, 1928). This discovery provided an explanation for how plants grow and allowed the exogenous application of this hormone to a multitude of species and tissues. Often, however, phenotypes from these exogenous assays were difficult to reconcile. For instance, auxin could promote root primordia formation yet repress root elongation whereas auxin could also inhibit bud activation yet promote tissue elongation (Skoog & Miller, 1957). Today these observations have been reconciled, but in the 1930s and 1940s, these results seemed contradictory. To address these challenges, several groups focused on using simplified experimental systems including

stem segments of *Nicotiana*. One such system, White's tobacco callus, was a cross between *Nicotiana glauca* and *Nicotiana langsdorffii* that could spontaneously form masses of undifferentiated cells and galls (White, 1939). Taking stem cuttings from these hybrids and culturing them on nutrient-rich media without hormones allowed an unlimited proliferation of growth with little differentiation (White, 1939). Skoog and colleagues used White's tobacco callus for early experiments but, due to difficulties obtaining cultures, developed instead a system that used stem cuttings from *Nicotiana tabacum* that underwent cell proliferation at sites of wounding on nutrient-rich media (Skoog & Tsui, 1948). Excising the inner pith tissues from such stem cuttings grew little but would undergo limited proliferation and extensive expansion in the presence of exogenously applied auxin (Jablonski & Skoog, 1954). Thus, by using a *Nicotiana* stem or pith-derived callus, Skoog and colleagues established a system that was relatively simplified and whose cell growth could be modified by the amounts of exogenous auxin treatment. However, auxin alone was not sufficient for the pith-derived callus to proliferate and instead a second factor was needed.

One hint about this cellular proliferation factor was that *Nicotiana* stem segments needed their vascular tissues to proliferate callus (Jablonski & Skoog, 1954). Several heterogeneous substances, including coconut milk, were found to induce cell proliferation but the first homogeneous chemical to show this effect was adenine (Skoog & Tsui, 1948). It behaved like a weak cytokinin (Amasino, 2005) but required high concentrations to promote cell proliferation (Skoog & Tsui, 1948). However, the results were clear from adenine and auxin treatments on *Nicotiana* stem cuttings: high adenine concentrations induced shoot formation whereas high auxin concentrations induced root formation (Skoog & Tsui, 1948). Treatments with both auxin and adenine caused cell proliferation but with neither root nor shoot formation (Skoog & Tsui, 1948). These results demonstrated that varying hormone concentrations could modify regenerative fates and determine organ identity. However, although these findings were published 9 years before the seminal 1957 paper, the 1948 paper did not reach the impact of Skoog and Miller's later work, perhaps due in part due to the challenges of working with adenine and the narrow concentration range at which this substance was active (Amasino, 2005).

In the 1940s and 1950s, groups were actively looking for robust chemical(s) that promoted cell proliferation. Through a combination of luck, hard work and carefully planned experiments, 6-furfurylaminopurine was found to promote cell proliferation at extremely low concentrations (Miller et al., 1955a, 1955b). This compound was renamed kinetin and was instrumental to revisiting the work done 9 years earlier with auxin and adenine. Using the in vitro *Nicotiana* system developed earlier and this recently discovered synthetic cytokinin (kinetin), Skoog and Miller employed these to make several fundamental discoveries. Firstly, they found that the presence of both auxin and cytokinin was required for cell proliferation; the presence of only one compound resulted in stem tissues growing poorly (Das et al., 1956; Skoog & Miller, 1957). Secondly, the effects of cytokinin and auxin were quantitative, that is, varying their concentrations would lead to vastly different morphological phenotypes. High levels of auxin and low levels of cytokinin promoted cell expansion and the formation of roots from callus tissues (Figure 1; Skoog & Miller, 1957). High levels of cytokinin and low levels of auxin promoted bud and shoot formation from callus. Similarly high levels of cytokinin and auxin differentiated neither shoot nor root, but instead favoured callus growth (Figure 1; Skoog & Miller, 1957). This effect is exemplified in Plate 4 of their 1957 paper (Figure 1a) when varying auxin and

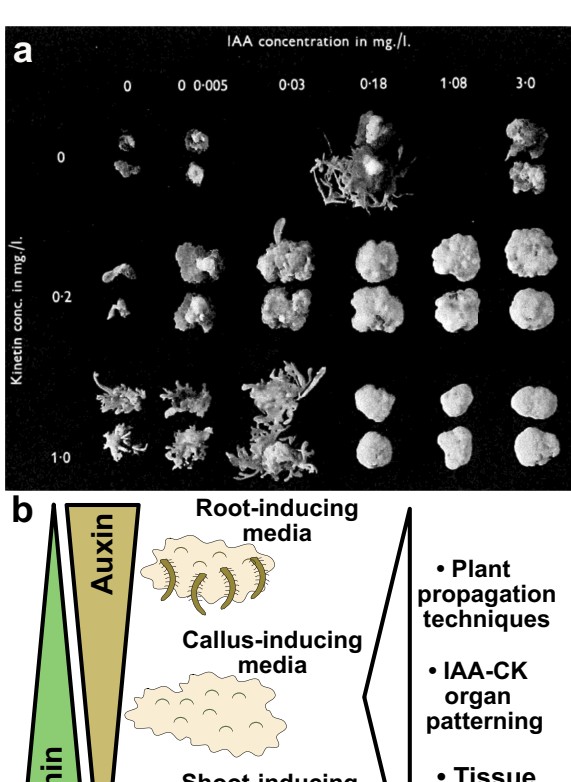

**Figure 1.** Findings and outcomes from Skoog and Miller. (a) A central finding from Skoog and Miller (1957) demonstrated that varying levels of auxin (indole-3-acetic acid; IAA) and cytokinin (kinetin) result in shoot formation or callus formation from *Nicotiana* stem segments. In this experiment, kinetin levels were too high to allow root formation. Image taken from Plate 4 of Skoog and Miller (1957). (b) Auxin and cytokinin are both required for organogenesis but also antagonise each other to promote either root or shoot formation. High auxin and low cytokinin, the basis for root-inducing media (RIM), promote root formation. High cytokinin and low auxin, the basis for shoot-inducing media (SIM), promote shoot formation. Equal levels of auxin and cytokinin, the basis for callus-inducing media (CIM), promote callus formation. These in vitro organ formation experiments have proven critical for understanding tissue regeneration and hormone-mediated organ patterning, whereas the use of CIM, SIM and RIM has been critical for both plant propagation and the regeneration of transgenic plants. Figure 1(a) © Cambridge University Press, 1957. Please note, the Open Access licence covering this article does not apply to this image.

cytokinin concentrations promoted callus or shoot growth (Skoog & Miller, 1957). In addition, there was a clear repressive effect of cytokinin upon root initiation and a clear repressive effect of auxin upon shoot formation (Skoog & Miller, 1957). Although these experiments were performed in *Nicotiana* stem segments and callus cultures, Skoog and Miller wisely speculated that these quantitative interactions between auxin and cytokinin might extend more broadly and cover all types of cellular growth and organ formation in plants (Skoog & Miller, 1957).

## 3. The implications of Skoog and Miller

Our mechanistic understanding of plant development and developmental plasticity has made quantum leaps since the work of Skoog and Miller in 1957. Here, I focus on three aspects that have led to their paper becoming a classic in plant biology and remaining a highly cited paper in the field.

The idea that qualitative interactions of growth hormones, for instance, the presence of a hormone regardless of amount, controlled growth was put into doubt by the findings of Skoog and Miller. Instead, their results indicated that small differences in hormone concentrations could modify the fate of organ development and growth. Two notable examples are mentioned in the 1957 paper that are still relevant and widely studied today. Firstly, they observed that auxin enhanced root formation while cytokinin repressed it (Skoog & Miller, 1957). Although unknown to Skoog and Miller, today we know that in dicots lateral roots emerge from the pericycle cells adjacent to the xylem. There, pericycle founder cell specification requires the activation of a local auxin response to promote lateral root formation (Dubrovsky et al., 2008). Treatment with cytokinin blocks lateral root initiation by perturbing the expression of the PIN auxin transport genes necessary for the formation of an auxin gradient in the lateral root founder cells (Laplaze et al., 2007). Auxin promotes PIN expression and stability (Adamowski & Friml, 2015), thus the balance between auxin and cytokinin influences PIN expression and ultimately either promotes or represses auxin response in the pericycle to determine lateral root formation. The second notable example from Skoog and Miller is that they observed cytokinin promoted bud formation whereas auxin repressed it (Skoog & Miller, 1957). Increasing cytokinin levels activates the expression of *WUSCHEL* and *SHOOT MERISTEMLESS*, positive regulators of shoot meristem cell fate (Gordon et al., 2009; Rupp et al., 1999). Overexpressing both *WUSCHEL* and *STM* is sufficient to form ectopic shoots (Gallois et al., 2002; Lenhard et al., 2002) suggesting that high cytokinin induces shoot formation via this pathway. How high auxin levels repress shoot formation is less clear. Auxin response factors suppress *SHOOT MERISTEMLESS* expression (Chung et al., 2019), whereas auxin also activates the cytokinin signalling inhibitor *AHP6* (Besnard et al., 2014). Thus, it is possible that high auxin levels inhibit shoot formation by decreasing *STM* expression and repressing cytokinin signalling.

A second outcome from Skoog and Miller's paper was to establish tissue regeneration that could be quantitatively manipulated. Although previous papers had developed in vitro methods to culture plant tissues, the combination of auxin and cytokinin facilitated the widespread study of tissue regeneration. Such an ability for somatic tissues to form new tissues and whole plants after wounding raised important questions for how such de novo organ formation might occur. One idea is that tissues like *Nicotiana* stems contain a subset of stem cells that can divide and differentiate to give rise to any cell-type, tissue, or whole organism (Birnbaum & Sanchez Alvarado, 2008). In contrast, another idea is that differentiated cells such as epidermis, mesophyll or root hairs can change their fate to form new cell types that can give rise to organs (Birnbaum & Sanchez Alvarado, 2008; Morinaka et al., 2023). Such fate changes could involve the change of one differentiated cell to another, a process known as trans-differentiation, or could involve the de-differentiation of cells followed by a re-differentiation process (Sugimoto et al., 2011). Likely both concepts are important, for instance, treatments with exogenous auxin specifically cause xylem pole pericycle cells to divide and give rise to lateral root-like meristems that transition to shoots after cytokinin treatment (Atta et al., 2009; Che et al., 2007; Gordon et al., 2007). Transcriptional analyses of these callus masses, whether derived from roots or aerial tissues, revealed that they had a similar identity to lateral root tips (Sugimoto et al., 2010). Thus, not all plant cells give rise to callus but instead these data indicated that only a subset of the pericycle cells adjacent to the xylem do and such masses are not undifferentiated but instead transcriptionally resemble root tips regardless of their tissue origin (Atta et al., 2009; Sugimoto et al., 2010). It seems appropriate then to consider xylem pole pericycle cells as having totipotency that, upon auxin and cytokinin treatment, transdifferentiate or de-differentiate/re-differentiate to other cell types that can give rise to whole plant regeneration.

Lastly, a third outcome from Skoog and Miller's paper was the establishment of in vitro conditions that allowed for efficient plant propagation. By taking cut tissues and placing them on varying concentrations of auxin and cytokinin, Skoog and Miller had invented callus inducing-media (CIM), shoot-inducing media (SIM) and root-inducing media (RIM) (Figure 1b). Such media were crucial for allowing organ regeneration from cut tissues, processes that have been developed and studied in hundreds of plant species. These media allowed the asexual propagation of plants, such as orchids, using micropropagation techniques. With the advent of transgenesis, T-DNAs could be transferred by *Agrobacterium* or biolistics directly into CIM-derived callus and transgenic plants regenerated using SIM and RIM techniques. Skoog and Miller in their 1957 paper observed the formation of roots and shoots directly from stem segments (Skoog & Miller, 1957), a process we refer to today as direct regeneration. However, work 30 years after Skoog and Miller revealed that such a process was less efficient than indirect regeneration. With indirect regeneration, plant segments were incubated on CIM for several days to gain regeneration competency, after which they were transferred to SIM to induce shoot formation (Feldmann & David Marks, 1986; Valvekens et al., 1988). Regeneration rates were substantially higher with indirect techniques and the time it took for shoots to form was dramatically reduced. Although Miller and Skoog established the fundamental concepts behind modern tissue culture, most regeneration protocols today involve modifications of these techniques to incorporate treatments and transfers between different inducing media.

## 4. Concluding remarks

The work from Skoog and Miller (1957) remains a classic and parts, including Plate 4 (Figure 1a), represent one of the most visually striking images from that generation of papers. Their paper was at the right time and place: in vitro techniques were already established, the effects of auxin upon growth were well known and a potent synthetic cytokinin had just been discovered two years prior by Miller, Skoog and colleagues. With a combination of well-planned experiments and visually striking outcomes, the work of Skoog and Miller will remain a classic and likely continue to be highly cited. Today, our understanding of phytohormones has expanded massively and we know that auxin and cytokinin work together in both synergistic and antagonistic functions at the cellular and tissue level. Furthermore, not only is hormone concentration relevant, but so too is the location of hormone response. This complexity confounded early developmental biologists, and today continues to present both a highly interesting research question but also a challenge. Perhaps we should take advice from Skoog and Miller and continue advocating for techniques, such as in vitro cultures, that simplify complex biological systems and help us dissect the complexities of plant growth and developmental plasticity.

**Competing interest.** The author declares none.

**Author contribution.** C.W.M. conceived and wrote the article.

**Funding statement.** Work in the Melnyk lab is supported by a Wallenberg Academy Fellowship (2016-0274) and a European Research Council starting grant (GRASP-805094).

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
