## [Reviewer Report]

Dear Editor,

Please see attached my classics review. I apologise this is coming so late!

Best regards,

Charles

---

## [Reviewer Report]

Dear Dr. Melnyk,

Thank you for submitting your manuscript “Quantitative regeneration: Skoog and Miller revisited” to QPB-Classics. I have now received all the referees‘ reports on the above manuscript, and have reached a decision. The referees’ comments are appended below, or you can access them online.

As you will see, both referees express considerable interest in your review article, but have some minor suggestions as well.

More specifically, the two reviewers found your paper concise, well written as it nicely summarizes the historical significance and the current relevance of a classic paper by Skoog and Miller on quantitative auxin/cytokinin applications. At the same time they have suggested that some technical terms need to be carefully selected [ For example: “totipotency”/“pluripotency.”; dedifferentiation/re-differentiation versus transdifferentiation]; they asked for some clarifications to be added where indicated within the text, and proposed some changes to the Figure, as well.

If you are able to revise the manuscript along the lines suggested, which I believe should mainly involve further text editing, I will be happy to receive a revised version of the manuscript for further consideration. Your revised paper will be re-reviewed by one or more of the original referees, and acceptance of your manuscript will depend on your addressing satisfactorily the reviewers' minor concerns.

Please attend to all of the reviewers‘ comments and ensure that you clearly highlight all changes made in the revised manuscript. Please avoid using ’Tracked changes‘ in Word files as these are lost in PDF conversion. I should be grateful if you would also provide a point-by-point response detailing how you have dealt with the points raised by the reviewers. Alternatively, if you do not agree with any of the referees’ criticisms or suggestions, please explain clearly why this is so.

Thank you again for submitting your nice review article to QPB, and I am looking forward to receiving your revised manuscript.

Sincerely,

Ali FERJANI

Quantitative Plant Biology, Associate Editor

---

## [Reviewer Report]

Dear Dr Hamant and Dr Ferjani,

Thanks for your comments and suggestions. I have revised the manuscript in light of the reviewer’s comments. Apologies this took so long. Hopefully the revised manuscript has addressed the previous issues and would be of interest to readers of Quantitative Biology.

Best regards,

Charles

---

## [Reviewer Report]

Dear Dr Melnyk,

Thank you for resubmitting your revised manuscript QPB-22-0018.R1 “Quantitative regeneration: Skoog and Miller revisited” to the QPB-Classics. Both reviewers and I found that you have adequately addressed all the minor points raised in the first version of the manuscript. Therefore, I am happy to recommend the publication of your invited article as is.

Thank you very much again for submitting your article to QPB, which we believe will represent a nice addition to the field of plant regenerative biology.

Sincerely,

Ali Ferjani

Associate Editor, Quantitative Plant Biology